# Digitalizing the community health information system improved women's retention on the maternal continuum of care pathway in northwest Ethiopia: A quasi-experimental study

**Tesfahun Hailemariam**[1,2*], **Asmamaw Atnafu**[3], **Lemma Derseh Gezie**[4],
**Jens Johan Kaasbøll**[5], **Jörn Klein**[6], **Binyam Tilahun**[1]

**1** Department of Health Informatics, Institute of Public Health, College of Medicine and Health Sciences, University of Gondar, Gondar, Ethiopia, **2** Department of Health Informatics, College of Health Sciences, Hawassa, Ethiopia, **3** Department of Health System and Policy, Institute of Public Health, College of Medicine and Health Sciences, University of Gondar, Gondar, Ethiopia, **4** Department of Epidemiology and Biostatistics, Institute of Public Health, College of Medicine and Health Sciences, University of Gondar, Gondar, Ethiopia, **5** Department of Informatics, University of Oslo, Oslo, Norway, **6** Department of Nursing and Health Sciences Campus Porsgrunn, University of South-Eastern Norway, Porsgrunn, Norway

* tesfahunhailemariam@gmail.com

## Abstract

### Background

Retaining women on the maternal continuum of care pathway remains a significant challenge for the healthcare system. Digitalizing primary healthcare system using community health workers is a key strategy to enhance maternal healthcare utilization in Ethiopia. However, the effectiveness of digitalizing community health system using frontline health system workers is uncertain.

### Objective

To determine the effect of electronic community health information system intervention on maternal continuum of care in northwest Ethiopia.

### Methods

A pre-post experimental study was conducted in Wogera district, northwest Ethiopia. Data were collected at household level from April to May 2022 including a total of 830 women with children with age less than one-year. A log-binomial logistic regression model was used to estimate the effect of electronic community health information system on the maternal continuum of care and its predictors. Relative risk with a 95% confidence interval was reported, with a p-value of <0.05 considered statistically significant.

**Data availability statement:** All relevant data are within the paper and its Supporting Information files.

**Funding:** This work would not be possible without the financial support from the Doris Duke Charitable Foundation (grant 2017187) through the University of Gondar.

**Competing interests:** The authors have declared that no competing interests exist.

## Results

The study revealed that 10.6% women were fully retained on the maternal continuum of care before the intervention while 32.5% after the intervention($p < 0.001$). Women in the intervention cluster had 3.12 times higher coverage of maternal continuum of care(ARR: 3.12,95%CI: 2.20–4.41). In addition, primary education(ARR: 1.54, 95%CI:1.14, 2.08), secondary and above education(ARR: 1.61, 95%CI:1.08,2.39); pregnancy intention (ARR: 1.67, 95%CI: 1.11,2.51), and women's autonomy in healthcare decision making (ARR: 2.02, 95%CI: 1.03, 3.97) were significantly associated with maternal continuum of care.

## Conclusions

Electronic community health information system improved maternal continuum of care. Maternal health service programs in rural should consider the implementation of electronic community health information system. Provision of women's education, prevention of unintended pregnancy, and enhancement of autonomy of women in healthcare decision making should be emphasized to improve maternal continuum of care.

## Background

Globally, an estimated 295000 maternal deaths occurred in 2017, yielding an overall maternal mortality ratio (MMR) of 211 maternal deaths per 100,000 live births globally, and more than nine out of ten of these deaths (94%) occurred in low- and middle-income countries (277,300 per 100,000 live births) [1,2]. Sub-Saharan Africa and southern Asia account for up to 86% (254000 per 100,000 live births). Sub-Saharan Africa alone accounts for 66% (196000 per 100,000 live births) [1,2]. Ethiopia has made remarkable gains in reducing maternal mortality over the past several decades, but the total number and rates of these deaths remained too high, and improving maternal health is remaining a major challenge in the health system of the country [3]. The current estimate shows that 401 preventable maternal deaths occur for every 100,000 live births in Ethiopia [1] linked with a low maternal continuum of care [4].

Many women die because of complications during and following pregnancy, and most of these complications are preventable [1]. Nearly 40% of women develop complications following delivery, and almost 15% of them encounter potentially life-threatening complications [5,6]. Poor antenatal and postnatal care attendance accounts for a substantial number of preventable maternal deaths [7]. Currently, countries are striving to reduce unwanted maternal deaths that occur during pregnancy and childbirth [8,9]. Moreover, it is the prime focus of sustainable development goals (SDGs) to ensure and sustain women's well-being [10].

The government of Ethiopia has been striving to promote women's good health through designing policies, strategies, and guidelines at the national and sub-national level and implementing for the betterment of maternal health programs aligned with the SDGs [10,11]. The long-and short-term training of midwifery nurses and the implementation of guidelines for clinical settings are some of the efforts of the government of Ethiopia to reduce the massive number of maternal deaths.

Even though subsequent maternal healthcare use is recommended to improve maternal health and birth outcomes, as well as to save the lives of women and newborns, healthcare utilization remains a significant challenge in resource-constrained settings in general and in Ethiopia in particular [3,12]. The global proportions of women attending at least one antenatal care visit in low-income, middle-income, and industrialized

regions were 70%, 68%, and 98%, respectively [13]. According to our review, maternal continuum of care in Ethiopia ranges from 9.7% [14] to 47% [15] and the pooled proportion of the maternal continuum of care in Ethiopia was found to be 25.51% [16] while 20.94% in Africa [17].

A study shows that nearly half of pregnant women (48%) in low and middle income countries do not receive antenatal care visit four or more [12], thus the, poor maternal healthcare utilization during pregnancy, childbirth, and after birth remains a leading cause of preventable maternal deaths [18].

In Ethiopia, about 1 out of 4 women (26%) do not receive any antenatal care (ANC) visits [19], and 17% had their first prenatal care visit before the fourth month of pregnancy [20]. Similarly, women who received at least four antenatal care visits were about 4 out of 9 (43%) pregnant women [19]. According to Ethiopian demographic health survey (EDHS) report, more than half (52%) of pregnant women do not attend facility delivery and the proportion of continuum of maternity care varies across regions in Ethiopia [19]. A study showed that both individual and cluster level factors are determining women's healthcare utilization [21].

Health management information system (HMIS) is a manual approach in healthcare data capturing and reporting system using standardized tools and procedures [22]. However, poor data quality, use, and and lack of informed-decision making are limitations in the paper based health system [23,24] which ultimately lead to poor performances of the healthcare system [25].

Digital health has become a solution to improving the health system in recent years [26–28] and has shown tremendous promise in increasing penetration and better functionality in the healthcare industry. Various studies suggested mobile health with low cost improves health services provided by community health workers [29], as it helps them to memorize the service components easily and improves their confidence to use [30]. Digital health supports community health workers to generate quality health data [31], improve healthcare delivery [32], help them to be effective in their job aids [24,33], and improve quality data generation, use, and service provision [34]. Moreover, introducing mobile health for routine activities in the primary healthcare system is relevant and recommended for improvement of maternal health services in low-income countries [32,34] and a remedy to strengthen routine health system data management and service provision [34].

Though there is growing evidence showing that mobile phone solutions improve health service delivery and health outcomes [26–28,35–38], the available evidence vindicates the fact that there is little information on the effect of digitizing CHIS on health service delivery [21,39,40].

The government of Ethiopia has envisioned digitalizing primary healthcare units through electronic community health information system (eCHIS) program, a high priority area of the information revolution. In eCHIS, HEWs deliver e-CHIS-based job-aid activities using tablet devices and enable them to record, use, and report household data and facilitate referral linkage of pregnant women within health centers and health posts. Though multiple studies have presented the effect of mobile health intervention on the improvement of maternal health services utilization [39,41,42], the role of trained community health workers (CHWs) involvement in maternal healthcare improvement using eCHIS is unknown. Understanding the effect of the eCHIS program on maternal health service could help the government make further scale-up efforts. Moreover, the study's findings inform policymakers to take intervention measures in improving maternal health program. Therefore, this study aimed to measure the effect of e-CHIS intervention on maternal continuum of care in northwest Ethiopia.

## Methods

### Study setting and period

The Amhara region is located between 8˚45′ N and 13˚45′ N latitude and 35˚46′ E and 40˚25′ E longitude in Northwest Ethiopia [43,44]. It is subdivided administratively into 12 zones and three town administrations, and the region has a total of 189 districts [45]. As of 2021, Ethio Telecom had launched 4G service in several towns in the region so as to enhance internet speed and accessibility [46]. According to EDHS 2019 [19], in the Amhara region, health facility delivery, percentage of women with a postnatal check during the first 2 days after birth, and percentage receiving antenatal care from a skilled provider were 54.2%, 39.8%, and 82.6%, respectively. This study was conducted at Wogera district to measure the effectiveness of the eCHIS intervention. The total population of the intervention district was 278942, with reproductive and pregnant women 64993 and 9401, respectively. The number of surviving infants at the time of the study was estimated at 8393 (Central Gondar Zone Health Bureau report, unpublished).

### Study design

A pre-post quasi-experimental study design was employed.

### Study population

Women who gave birth within the last 12 months preceding the survey and were permanent residents of the intervention district were involved in the study.

### Sample size and sampling procedure

To determine the effect of eCHIS intervention on the maternal continuum of care, the sample size was calculated using GPower 3.1.9.4 version, considering the proportion of anatenatacal care visits four and above, which was reported at baseline paper, p1 = 39.6% [47]. Considering the effect size that the intervention could bring change on the interest of outcome, 10% change at the intervention district was included during the sample size calculation, which gives the proportion two (p2 = 49.6%) [29]. 5% level of significance, 80% power, and one to one ratio between baseline and endlin were considered Thus, baseline (n1) and endline (n2) of n1:n2 = 1:1, the total sample size was estimated at: n1 = n2 = 404. This sample size estimate is closer to that of the baseline study, which was 415, and to maintain better power for the study, the same sample size, i.e., 415 was employed for the endline study, giving a total sample size of 830, which considers the two samples.

We used a multi-stage sampling technique to reach the study participants in the study cluster (Fig 1). Simple random sampling technique was applied to select the health centers in the intervention district, and a systematic random sampling approach was employed to select the study particiapnts. To get the study sampling frame, we used a family folder, which was available at primary healthcare units at the heatlh post level and used by HEWs to document pregnant women in the cathment health post [48]. The sampling interval was obtained considering the total sample size estimated (830) and the eligible source proptuion at the district (8393). The first study participant (the 2nd mother) was selected by a simple random sampling technique, and all other mothers were selected systematically by taking every 10th mother in the frame. Thus, the study particiapnts were selected systematically using the samling interval. During the data collection for women who were selected and not avaialbale at their house, we gave wating time two to five days to get them back and include in the study.

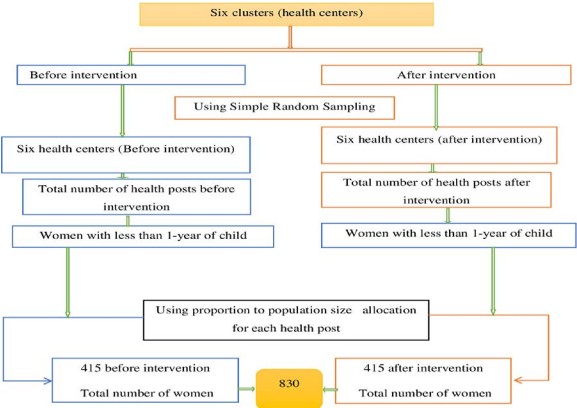

**Fig 1. Schematic presentation of sampling procedure for study participants before and after the intervention in Wogera District, 2022.**

## Intervention description

To tackle the challenges related to quality health data production, use, and service provision, the government of Ethiopia has taken the initiative to digitize the existing paper-based CHIS through the e-CHIS system. The Amhara regional state health bureau, along with the University of Gondar (UoG), carried out the eCHIS implementation in the Wogera district of northwest Ethiopia. The intervention was skill-oriented training for the implementers on mobile-based community health information system application followed by woreda led household registration, tablet usage guideline provision, technical support and mentoring, and periodical communications as detailed in (Fig 2).

During intervention, the following strategies were employed (Fig 2). Provision of mentorship and technical assistance: three supporting team members were assigned and provided technical assistance for implementers biweekly with a local mentor and every two weeks by supporting university throughout the intervention period. In addition, one health information

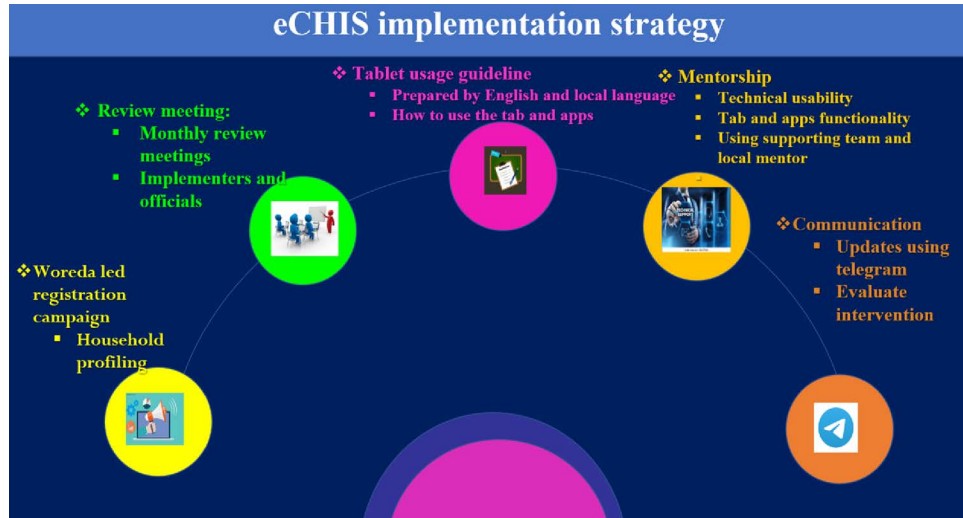

**Fig 2. Electronic community health information system implementation strategies in northwest Ethiopia, 2022.**

technician (HIT) technician (local mentor) was assigned to provide mentorship and solve eCHIS related problems during implementation.

Communication: Under each cluster, there was a focal person and a midwifery nurse for information exchange and programmatic assistance. A group Telegram channel was created and used as a platform to share information daily in relation to eCHIS progress.

Tablet usage policy: the tablet usage policy was developed both in English and local language by the technical support team from UoG to enable all users to use the tablet devices properly for e-CHIS activities and to ensure the functionality of the system during implementation.

Review meeting: during the entire period, monthly review meetings were conducted in connection with eCHIS implementation amongst the head of the woreda health office, the woreda health office planning officer, the head of the district administration, the head of the zonal health bureau, the head of Amhara regional health bureau planning, and the technical supporting team from UoG.

During the implementation, health extension workers (HEWs) in the district were trained and performed household and pregnant women registration using customized electronic forms. The HEWs used the tablet for community health data management and service provision using the HEWs application. The pregnant women in the intervention cluster were linked with the women's development army using the eCHIS system. Health centers and respective health posts under eCHIS implementation clusters were considered to determine the effect of digitalizing community health information system on the maternal continuum of care (Fig 3). The usual activities in the comparator district were HEWs deliver routine activities using community health information system (CHIS) folders in the community. We compared the influence of the digital component, eCHIS, on the maternal health service outcomes before and after the implementation of eCHIS.

## Outcome

The dependent variable of the study was maternal continuum of care. Maternal continuum of care was categorized as 'yes' when a woman completes all the required maternal healthcare utilization from antenatal care visit to postnatal care visit within 48 hours after birth. Those women who did not fulfill complete maternal continuum of care components were categorized as 'no' [49].

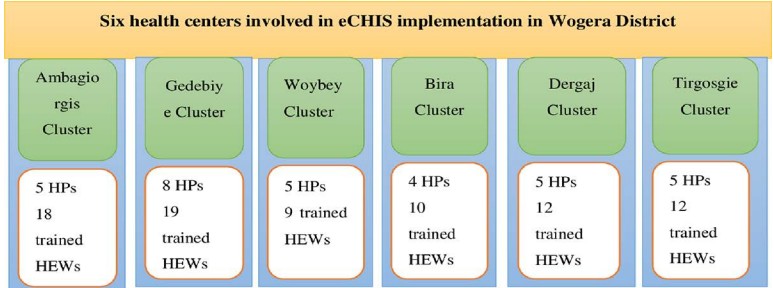

**Fig 3. Intervention clusters and HEWs involved in eCHIS implementation in Wogera, district northwest Ethiopia, 2022.**

## Exposure

Electronic community information system and multiple variables such as age of the respondents; marital status of the respondents; education status of women; education status of their partners; pregnancy intention; membership of community-based health insurance; HEWs' home visits during pregnancy; women's autonomy in healthcare and decision-making at the household; family planning, parity, and complications during pregnancy period were included in the study.

## Data collection instrument, procedures, and quality

The data collection tool was developed in English and translated into the local language (Amharic). Experts in the area were invited to review the relevance of each question in the instrument. Thus, the tool was revised according to the experts' views and was piloted out of the selected clusters and validated before data collection. We employed face and content validity; thus, the rating points were 0.98 and 0.81, respectively. The data collectors and supervisors were trained on the data collection tools and procedures. The data were collected via face to face using an interviewer-administered questionnaire.

## Operational definition

Digitalization refers to the use of digital devices to reduce costs, improve business productivity, and transform existing practices [50]. eCHIS digitizes the family folder and community health information system content into a mobile platform for use by HEWs.

## Data management and statistical analysis

Data cleaning and management were employed using Stata 14. Frequency, percentages of the study variables were computed against the outcome of the interest, maternal continuum of care. A bivariable log-binomial logistic regression model was employed to identify the candidate variables for multivariable log-binomial logistic regression analysis at p-value $< 0.25$. Multivariable log-binomial logistic regression analysis was performed by estimating the risk ratios with a 95% confidence interval. A p-value of $< 0.05$ was considered to identify a significant predictor of maternal continuum of care. Multicollinearity was checked using a variance inflation factor. The cutoff point of 10 was used to identify the presence of multicollinearity among predictors [51].

## Ethical considerations

Verbal consent was obtained, and participants were informed about the objective and importance of the study, procedure and duration; risk and discomfort, benefits of participating in the study, confidentiality, and the right to refuse or withdraw during data collection. Study approval and ethical clearance were obtained from the University of Gondar ethical review board (R.NO. V/P/RCS/05/2020). A formal letter of approval was obtained from the Amhara national regional state health bureau and the Central Gondar zonal health department. Informed consent was obtained from all subjects and/or their legal guardian(s). For participants aged $< 18$, verbal informed consent was obtained from their parents and assent was obtained from the minor/participant.

## Results

### Socio-demographic and reproductive characteristics

A total of 830 women participated in the analysis (n1 = n2 = 415) with a response rate of 100%. The mean (standard deviation, SD) age of participants were 29 and 6.7, respectively,

and the minimum and maximum ages were 15 and 49 years, respectively. About 261 (31.4%) and 107 (12.9%) of women attended primary, secondary and above education, respectively. The participants who had membership in community-based health insurance were 551 (66.4%). About 650 (78.3%) of women had pregnancy intentions for their current child, and 16.2% had autonomy in healthcare seeking and decision-making. The study showed that about 400 (48.2%) of the participants were visited by HEWs during pregnancy period (Table 1).

The study showed that in pre-post comparison, in the intervention district, 44(10.6%) and 135(32.5%) (p < .001) of women were retained fully on the maternal continuum of care (Fig 4).

## Maternal continuum of care

The finding showed that maternal continuum of care in the intervention group increased by 21.9 percentage points compared to comparator group (Table 2).

## Multivariable log-binomial logistic regression analysis

The study showed that maternal education, pregnancy intention, and the autonomy of women for healthcare and decision-making were significantly associated with the maternal

**Table 1.  Socio-demographic and reproductive characteristics of the study participants in northwest Ethiopia before and after intervention, 2022 (n = 830).**

| Variables | Category | Before intervention n (%) | After intervention n (%) | Total n (%) |
|---|---|---|---|---|
| Age(years) | 15–24 | 121(14.6%) | 102(12.3%) | 223(26.9) |
| | 25–34 | 197(23.7%) | 195(23.5%) | 392(47.2) |
| | 35 and above | 97(11.7%) | 118(14.2%) | 215(25.9) |
| Marital status | Married | 407(49.0%) | 392(47.2%) | 799(96.2) |
| | Others | 8(1.0%) | 23(2.8%) | 31(3.8) |
| Mother's education | Did not attend education | 244(29.4%) | 218(26.3%) | 462(55.7) |
| | Primary education | 109(13.1%) | 152(18.3%) | 261(31.4) |
| | Secondary and above | 62(7.5%) | 45(5.4%) | 107(12.9) |
| Spouse's education | Did not attend education | 261(31.4%) | 223(26.9%) | 484(58.3) |
| | Primary education | 110(13.3%) | 166(20.0%) | 276(33.3) |
| | Secondary and above | 44(5.3%) | 26(3.1%) | 70(8.4) |
| Pregnancy intention | Yes | 328(39.5%) | 322(38.8%) | 650(78.3) |
| | No | 87(10.5%) | 93(11.2%) | 180(21.7) |
| Being insured | Yes | 198(23.9%) | 353(42.5%) | 551(66.4) |
| | No | 217(26.1%) | 62(7.5%) | 279(33.6) |
| Role of women in decision making | Husband | 4(0.5%) | 46(5.5%) | 50(6) |
| | Myself | 44(5.3%) | 90(10.9%) | 134(16.2) |
| | Jointly (husband and wife) | 367(44.2%) | 278(33.6%) | 645(77.8) |
| Parity | Para one | 99(11.9%) | 92(11.1%) | 191(23) |
| | Multipara | 316(38.1%) | 323(38.9%) | 639(77) |
| History of pregnancy related complications | Yes | 72(8.7%) | 43(5.2%) | 115(13.9) |
| | No | 343(41.3%) | 372(44.8%) | 715(86.1) |
| Family planning | Yes | 336(40.5) | 359(43.3) | 695(83.7) |
| | No | 79(9.5) | 56(6.7) | 135(16.3) |

Others: divorced, widow, and separated.

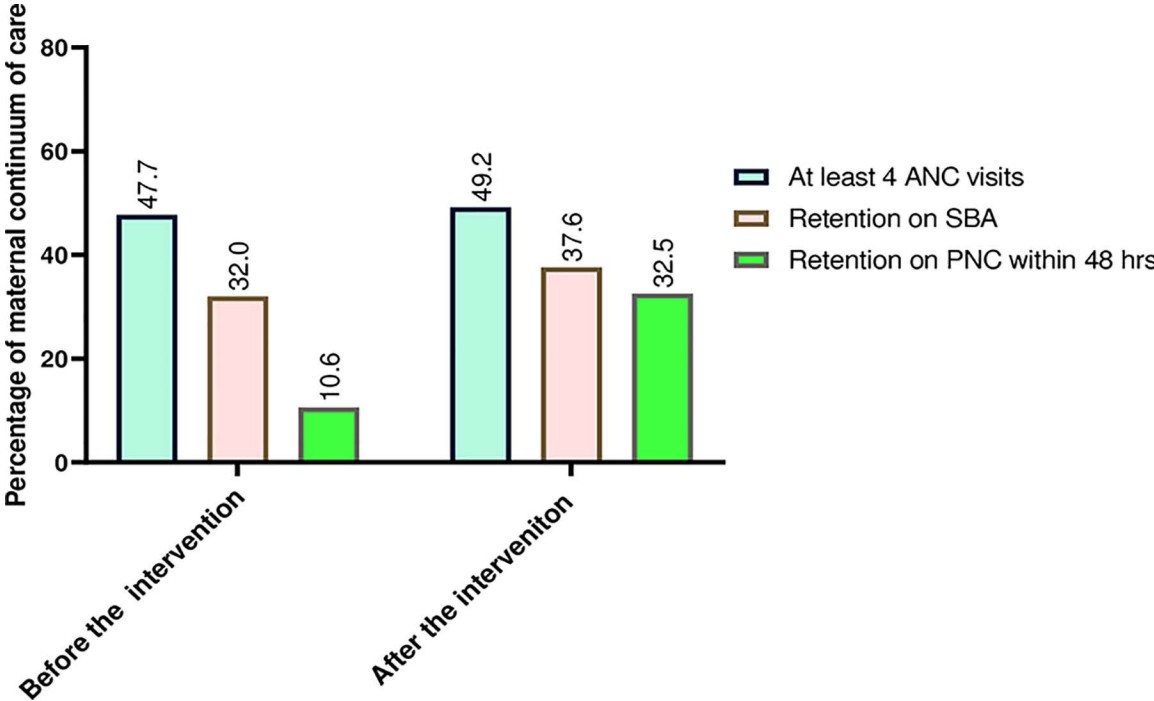

**Fig 4.** **Maternal health service utilization before and after the intervention of eCHIS in Wogera district, northwest Ethiopia, 2022 (n1 = n2 = 415).**

**Table 2.** **The effect of eCHIS intervention on maternal continuum of care in the intervention district, northwest Ethiopia, 2022.**

| Variables | Category | Maternal continuum of care | | Effect size (95%CI) |
|---|---|---|---|---|
| | | Yes n(%) | No n(%) | |
| Exposure to eCHIS intervention | Yes | 135(16.3) | 280(33.7) | 21.9% (18.9–24.6)* |
| | No | 44(5.3) | 371(44.7) | 1 |

* Stands for p-value < 0.001.

continuum of care. After controlling for confounders, women in the intervention group had (ARR: 3.12, 95% CI: 2.20–4.41) a higher chance of completing the maternal continuum of care compared to women in the comparator group. Women with a primary education level were (ARR: 1.54, 95%CI: 1.14–2.08) more likely to complete the maternal continuum of care than women with no formal education. Similarly, women with secondary and above education levels were (ARR: 1.61, 95%CI: 1.08–2.39) more likely to complete the maternal continuum of care as compared to their counterparts.

Women who were intended to be pregnant with the current child were 1.67 (ARR: 1.67, 95%CI: 1.11, 2.51) more likely to complete the maternal continuum of care than those who had no intention of pregnancy. Moreover, women with decision-making power at household level had a 2.02 (ARR: 2.02, 95%CI: 1.03–3.97) higher chance of completing the maternal continuum of care as compared to women who had no decision-making power for healthcare seeking at household level (Table 3).

**Table 3. A multivariable log-binomial logistic regression model for predictors of women's completion of maternal continuum of care in northwest Ethiopia, 2022 (N = 830).**

| Variables | Category | Maternal continuum of care | | Unadjusted RR (95%CI) | Adjusted RR (95%CI) | P-value |
|---|---|---|---|---|---|---|
| | | Yes n(%) | No n(%) | | | |
| Exposure to eCHIS intervention | Yes | 135(16.3) | 280(33.7) | 3.07(2.25,4.19) | 3.12(2.20,4.41) | 0.000 |
| | No | 44(5.3) | 371(44.7) | 1 | 1 | |
| Age group of respondents (years) | 15–24 | 55(6.6) | 168(20.2) | 1 | 1 | – |
| | 25–34 | 86(10.4) | 306(36.9) | 0.89 (0.66,1.19) | 0.99(0.72,1.36) | 0.723 |
| | 35 and above | 38(4.6) | 177(21.3) | 0.72(0.96,1.04) | 1.06(0.69,1.62) | 0.698 |
| Mother's education | Did not attend education | 71(8.6) | 391(47.1) | 1 | 1 | – |
| | Primary education | 75(9) | 186(22.4) | 1.87(1.40,2.49) | 1.54(1.14,2.08) | 0.005 |
| | Secondary and above | 33(4) | 74(8.9) | 2.01(1.41,2.86) | 1.61(1.08,2.39) | 0,019 |
| Spouse's education | Did not attend education | 85(10.2) | 399(48.1) | 1 | 1 | – |
| | Primary education | 74(8.9) | 202(24.3) | 1.53(1.16,2.01) | 1.13(0.85,1.49) | 0.394 |
| | Secondary and above | 50(6) | 20(2.4) | 1.63(1.07,2.47) | 1.26(0.84,1.88) | 0.269 |
| Pregnancy intended | Yes | 156(18.8) | 494(59.5) | 1.87(1.25,2.82) | 1.67(1.11,2.51) | 0.013 |
| | No | 23(2.8) | 157(18.9) | 1 | 1 | |
| Being insured | Yes | 138(16.6) | 413(49.8) | 1.70(1.24,2.34) | 0.90(0.65,1.26) | 0.547 |
| | No | 41(4.9) | 238(28.7) | 1 | 1 | |
| History of pregnancy related complications | Yes | 15(1.8) | 100(12) | 0.57(0.35,0.93) | 0.71(0.44,1.14) | 0.152 |
| | No | 164(19.8) | 551(66.4) | 1 | 1 | |
| Parity | Para one | 54(6.5) | 137(16.5) | 1.45(1.09,1.90) | 1.16(0.83,1.61) | 0.396 |
| | Multi para | 125(15.1) | 514(61.9) | 1 | 1 | |
| Role of women in decision making | Husband | 8(1) | 42(5.1) | 1 | 1 | – |
| | Myself | 42(5.1) | 92(11.1) | 1.96(0.99,3.88) | 2.02(1.03,3.97) | 0.042 |
| | Jointly (husband and wife) | 129(15.6) | 516(62.2) | 1.25(0.65,2.40) | 1.68(0.87,3.21) | 0.121 |
| HEWs Home visiting | Yes | 99(11.9) | 301(36.3) | 1.33(1.20,1.73) | 0.91(0.68,1.83) | 0.480 |
| | No | 80(9.6) | 350(42.2) | 1 | 1 | |
| Family planning | Yes | 158(19) | 537(64.7) | 1.46(0.96,2.22) | 1.45(0.98,2.15) | 0.063 |
| | No | 21(2.6) | 114(13.7) | 1 | 1 | |

## Discussion

Nearly one in every nine women completes the maternal continuum of care at baseline, while one in every three women completes the maternal continuum of care at the study endpoint. Exposure to intervention, women's education, pregnancy intention, and women's autonomy in healthcare and decision-making were associated with the completion of the maternal continuum of care. Our study indicated that women in the intervention group had a higher chance of completing the maternal continuum of care compared to women in the comparator group.

The findings agreed with past studies showing that using mobile health involving community health workers has impacted maternal healthcare utilization [32,34]. The first possible reason for the improvement of the maternal continuum of care in our intervention district could be the fact that HEWs, focal persons, and a midwifery nurse provided maternal health services in integration in which pregnant women had a chance to get registered in the application and receive maternal health service.

A digitized health extension program (HEP) program could enable HEWs workers to provide maternal health services and link pregnant women with health facilities. According to the literature, mobile health interventions bring a positive effect on antenatal care, skilled

birth attendance, and postnatal care improvement [40], and empowering HEWs [52] yields improved maternal health service utilization [34].

The other possible reason could be the fact that eCHIS interventions might have increased the chance of women to access health education and maternity services that could enable them to adhere to maternal health service utilization. Studies show that mobile health increases adherence to health service utilization, develops health seeking behavior, decreases the chance of missing health care appointments, and enhances client and care provider relationships [53,54].

The other reason could be the connection of women in women development army (WDA), one of the eCHIS functions that could enable women to be agents of health and well-being in the community. The WDA is a structural arrangement that involves women's development team comprising 1-to-5 connections in the community. In WDA, community members get linked with the primary health care system and easily access actionable health messages [55] that have the ability to increase the efficiency of HEWs to refer more women with primary healthcare for maternity care [56].

The finding showed that women's education was significantly associated with the maternal continuum of care. The current finding was complemented by the studies conducted in Ethiopia [15,49,57,58] and elsewhere [59–63].

The association between the completion of the maternal continuum of care and education could be explained by the fact that educated women could easily understand and take action that was delivered from different sources, such as health messages from healthcare providers and different media. In addition, the more a woman educated, the more she could be self-determinant about herself, particularly in seeking healthcare and medical examinations, and could be the one to choose on her own without waiting for others to decide on her behalf.

Moreover, cultural influences that could hinder women from getting healthcare services might not affect an educated woman, which is a common influencing factor for women not to seek healthcare services in developing countries [64]. A study showed that obeying family members or those who have a role in family decision-making is considered a norm or culture, which has potential to harm women's health-seeking behavior [65]. Female education is a proxy factor to improve maternal healthcare utilization [66], and the more a woman becomes educated, the more she is able to seek healthcare from a healthcare provider [67].

Women who had an intention for their pregnancy were more likely to be retained on the maternal continuum of care paths as compared to women who had no intention of pregnancy for their current child. This finding is similar to the study findings from Ethiopia [68] and Bangladesh [69]. The possible reason for retention of women on the maternal continuum of care among women who had intended pregnancy could be emotional and psychological readiness among those who were intended to be pregnant more than their counterparts. Those who were not ready to be pregnant may not give emphasis to seeking care from healthcare providers, and even they may not give care for their pregnancy [70]. A study indicated that women who have intention to their pregnancy were more likely to receive maternal healthcare than who have no intention to their pregnancy [71].

Moreover, our study showed that women with the ability to decide or who had a role in making a decision at the household level for healthcare were nearly twofold greater in terms of completing the maternal continuum of care than women whose husbands had a role in making decisions for healthcare-seeking. Our finding is consistent with studies conducted in Ethiopia [15,72] and abroad [73]. This can be justified by the notion that the more the woman is autonomous in decision-making, the more she will be able to increase bargaining power with her husband to get maternity care [72]. A study, in this regard, showed that women's autonomy in making personal decisions increases the level of healthcare utilization [74].

Another possible explanation is that the Ethiopian government has been envisioning promoting gender equality through the implementation of different policies and strategies, such as providing training on women's empowerment for healthcare decision-making to achieve the 2030 SDG agendas [10,11].

## Limitations and strengths

The study could have limitations related to unobserved sources and the absence of a control group, as this study was conducted using a pre-post quasi-experimental study design. Those women who had previous complications related to their pregnancy were not excluded; this could bring social desirability bias. Recall bias may exist in the current study as the participants were asked about the previous healthcare utilization exposure. In this study, we applied the probability sampling technique to reduce selection bias that could arise during study cluster and participant selection. To minimize the confounding effect, potential variables were included, and different statistical approaches were applied during analsysis by restricting the eligibility criteria.

## Conclusion

One in every three women completed the maternal continuum of care at the study endpoint, which was one in every nine women at baseline. The findings suggest that eCHIS intervention is effective in improving the maternal continuum of care. Women's education, pregnancy intention, and women's autonomy in healthcare and decision-making for healthcare were factors influenced completion of the maternal continuum of care. Therefore, maternal health service programs in rural areas should consider implementation of eCHIS, provision of women's education, prevention of unintended pregnancy, and enhancement of the autonomy of women in healthcare and decision-making to improve the maternal continuum of care.

## Supporting information

**S1 Data. Dataset of Digitalizing the community health information system improved women's retention on the maternal continuum of care pathway in northwest Ethiopia: A quasi-experimental study.**
(DTA)

## Acknowledgments

We thank the Amhara national regional state health bureau, Central Gondar zone health department, Wogera district health office, University of Gondar Comprehensive Specialized Hospital, and HEWs for their provision of necessary information and support during data collection. Our gratitude also goes to the study participants, data collectors, and supervisors who took part in the study.

## Author contributions

**Conceptualization:** Tesfahun Hailemariam, Asmamaw Atnafu, Lemma Dersehu Gezie, Jens Johan Kaasbøll, Jörn Klein.

**Data curation:** Tesfahun Hailemariam, Asmamaw Atnafu, Lemma Dersehu Gezie, Jens Johan Kaasbøll, Jörn Klein, Binyam Tilahun.

**Formal analysis:** Tesfahun Hailemariam, Asmamaw Atnafu, Lemma Dersehu Gezie, Jens Johan Kaasbøll, Jörn Klein, Binyam Tilahun.

**Investigation:** Tesfahun Hailemariam, Asmamaw Atnafu, Lemma Derseh Gezie, Jens Johan Kaasbøll, Jörn Klein, Binyam Tilahun.

**Methodology:** Tesfahun Hailemariam, Asmamaw Atnafu, Lemma Derseh Gezie, Jens Johan Kaasbøll, Jörn Klein, Binyam Tilahun.

**Project administration:** Tesfahun Hailemariam, Asmamaw Atnafu, Lemma Derseh Gezie, Jens Johan Kaasbøll, Jörn Klein, Binyam Tilahun.

**Resources:** Tesfahun Hailemariam, Asmamaw Atnafu, Lemma Derseh Gezie, Jens Johan Kaasbøll, Jörn Klein, Binyam Tilahun.

**Software:** Tesfahun Hailemariam.

**Supervision:** Tesfahun Hailemariam, Asmamaw Atnafu, Lemma Derseh Gezie, Jens Johan Kaasbøll, Jörn Klein, Binyam Tilahun.

**Validation:** Tesfahun Hailemariam.

**Writing – original draft:** Tesfahun Hailemariam.

**Writing – review & editing:** Tesfahun Hailemariam, Asmamaw Atnafu, Lemma Derseh Gezie, Jens Johan Kaasbøll, Jörn Klein, Binyam Tilahun.

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
