## [Decision Letter · Decision Letter 0]

29 Aug 2024

PONE-D-24-09283Impact of digitalizing the community health information system on women’s retention in the maternal continuum of care pathway in northwest EthiopiaPLOS ONE

Dear Dr. Hailemariam,

Thank you for submitting your manuscript to PLOS ONE. After careful consideration, we feel that it has merit but does not fully meet PLOS ONE’s publication criteria as it currently stands. Therefore, we invite you to submit a revised version of the manuscript that addresses the points raised during the review process.

We look forward to receiving your revised manuscript.

Kind regards,

Amanuel Yoseph, MPH

Academic Editor

PLOS ONE

“We thank the Amhara national regional state health bureau, Central Gondar zone health department, Wogera district health office, University of Gondar Comprehensive Specialized Hospital, and HEWs for their provision of necessary information and support during data collection. Our gratitude also goes to the study participants, data collectors, and supervisors who took part in the study. This work would not be possible without the financial support from the Doris Duke Charitable Foundation through the University of Gondar. “

6. We note that Figure 1 in your submission contain [map/satellite] images which may be copyrighted. All PLOS content is published under the Creative Commons Attribution License (CC BY 4.0), which means that the manuscript, images, and Supporting Information files will be freely available online, and any third party is permitted to access, download, copy, distribute, and use these materials in any way, even commercially, with proper attribution. For these reasons, we cannot publish previously copyrighted maps or satellite images created using proprietary data, such as Google software (Google Maps, Street View, and Earth). For more information, see our copyright guidelines: http://journals.plos.org/plosone/s/licenses-and-copyright.

Additional Editor Comments:

I critically reviewed your article entitled “Impact of digitalizing the community health information system on women’s retention in the maternal continuum of care pathway in northwest Ethiopia” which has the potential to add to the existing body of scientific knowledge, particularly in developing countries. However, there are some limitations in your article that need addressing before publication.

1. There are several grammatical and typological errors that authors need to carefully review.

2. Authors should extensively format manuscripts based on PLOS ONE journal style, including file naming. Avoid unnecessary italicizing and capitalization throughout the manuscript.

3. Make sure that your reference contains all the necessary details and PLOS ONE style.

Decision: Major revision

Reviewers' comments:

Reviewer's Responses to Questions

**Comments to the Author**

1. Is the manuscript technically sound, and do the data support the conclusions?

Reviewer #1: Yes

Reviewer #2: Partly

Reviewer #3: Partly

Reviewer #4: Yes

2. Has the statistical analysis been performed appropriately and rigorously? 

Reviewer #1: Yes

Reviewer #2: Yes

Reviewer #3: Yes

Reviewer #4: Yes

3. Have the authors made all data underlying the findings in their manuscript fully available?

Reviewer #1: No

Reviewer #2: Yes

Reviewer #3: Yes

Reviewer #4: Yes

4. Is the manuscript presented in an intelligible fashion and written in standard English?

Reviewer #1: Yes

Reviewer #2: Yes

Reviewer #3: No

Reviewer #4: Yes

5. Review Comments to the Author

Reviewer #1: Dear author, thank you for your effort to assess the impact of digitalizing the community health information system on women’s retention in the maternal continuum of care pathway in northwest Ethiopia. The manuscript has addressed an important topic, but incorporating the following comment can make the manuscript more understandable and repeatable.

Please give a line number; it makes it easy to comment and also helps determine whether the comments were addressed.

Some of the paragraphs lack references, and even all paragraphs are cited from a single paragraph. This shows adequate and relevant literature was not consulted. The manuscript can benefit from adequate analysis and synthesis of the literature in the field.

Please introduce each abbreviation in the first appearance in the manuscript. E.g. MMR, on the first line of page 3.

In the method section, please elaborate on the description of the study area. Include the regional state, zone, and other key socio demographic and economic features of the area. Also, the health service coverage, other infrastructure, including telecommunications .

It's more convenient to write the study design and population separately. It is wise to be specific while writing the study design, usually with a simple sentence. However, a description of the population may require a paragraph.

In the description of the intervention, the implementers and the implementation process were overemphasized (the government, the Amhara RHB).

Please describe the formula and the software used for sample size calculation. It's more appropriate if the sample size calculation and sampling technique come after the population section.

Please make sure the definition given to the outcome variable is clear and measurable. Also, in the references cited, the outcome was measured in the same way.

Do all the variables listed in the exposure variable were exposure? For the outcome. Do they have levels?

Clearly describe the types of data collection methods, did you prepare them or were they already used in similar studies previously? Clarify if the instrument is standard.

Also, write the last sentence of the data collection instrument section, as this sentence talks about data quality control and not about data collection tools.

On the ethical consideration part; why did you prefer to obtain verbal consent?

On the result part, please add the response rate.

What was your reference to categorize the age of the participant into three categories?

Page 10 of the maternal continuum of care before intervention was not clearly mentioned.

In table 4, log the binomial logistic regression model, add the p value.

On the discussion part, the paragraphs were too long. It is advisable to finalize the paragraph in 5–6 lines addressing the same idea.

Give a heading for the limitations of the study.

Reviewer #2: Comments Authors

The study effectively demonstrates the positive impact of the electronic Community Health Information System (eCHIS) on maternal health service utilization. However, there is a notable disconnect between the study's introduction, which emphasizes the need for research on the role of trained Community Health Workers (CHWs) in utilizing eCHIS, and the results and conclusions, which do not sufficiently address this focus. To improve coherence, I recommend that the authors either integrate findings related to the CHWs' contributions into the results and conclusions or adjust the introduction to better align with the broader scope of the analysis presented.

Furthermore, it is important to acknowledge the absence of a control group, as this limitation raises concerns that the observed changes might not solely result from the eCHIS implementation, affecting the study's generalizability. To enhance transparency and reproducibility please specify the exact sampling interval used in the systematic random sampling process.

Lastly, I suggest updating outdated references to include more recent research, which would reinforce the study's relevance and academic rigor.

Reviewer #3: Medium revisions: The manuscript is strong and contributes valuable insights into maternal health care in Ethiopia. Medium revisions to improve language clarity and provide additional details in the methods and discussion sections will enhance the manuscript's overall quality.

Reviewer #4: 1.The abstract should not exceed 300 words

2.The study design and study population are not well described.

3.Did you differentiate when to use simple random sampling and systematic random sampling?

4.What is wrong if we use simple random sampling to select study participants?

6. PLOS authors have the option to publish the peer review history of their article (what does this mean? ). If published, this will include your full peer review and any attached files.

**Do you want your identity to be public for this peer review?** For information about this choice, including consent withdrawal, please see our Privacy Policy .

Reviewer #1: **Yes: ** Dawit Getachew

Reviewer #2: No

Reviewer #3: No

Reviewer #4: No

---

## [Author Response · Author response to Decision Letter 0]

19 Sep 2024

Thank you dear respected reviewers. We are so greateful having you and getting your very valuable comments on our manuscirpt. We benifited a lot from your comments and your feedback were very helpful to improve our work. May be if we missed anthing during the reviosn, would you inform us to revise again and work on the mansuciprt, please? Thank you so much!

---

## [Decision Letter · Decision Letter 1]

29 Jan 2025

PONE-D-24-09283R1Digitalizing the community health information system improved women’s retention on the maternal continuum of care pathway in northwest Ethiopia: a quasi-experimental studyPLOS ONE Dear Dr. Hailemariam,

Thank you for submitting your manuscript to PLOS ONE. After careful consideration, we feel that it has merit but does not fully meet PLOS ONE’s publication criteria as it currently stands. Therefore, we invite you to submit a revised version of the manuscript that addresses the points raised during the review process.

If applicable, we recommend that you deposit your laboratory protocols in protocols.io to enhance the reproducibility of your results. Protocols.io assigns your protocol its own identifier (DOI) so that it can be cited independently in the future. For instructions, see: https://journals.plos.org/plosone/s/submission-guidelines#loc-laboratory-protocols . Additionally, PLOS ONE offers an option for publishing peer-reviewed Lab Protocol articles, which describe protocols hosted on protocols.io. Read more information on sharing protocols at https://plos.org/protocols?utm_medium=editorial-email&utm_source=authorletters&utm_campaign=protocols .

We look forward to receiving your revised manuscript.

Kind regards,

Birhan Tsegaw Taye

Academic Editor

PLOS ONE

Journal Requirements: Authors should extensively format manuscripts based on PLOS ONE journal style, including file naming. Avoid unnecessary italicizing and capitalization throughout the manuscript.

Additional Editor Comments:

I critically reviewed your article entitled “Digitalizing the community health information system improved women’s retention on the maternal continuum of care pathway in northwest Ethiopia: a quasi-experimental study,” which has the potential to add to the existing body of scientific knowledge, particularly in developing countries. However, there are some limitations in your article that need addressing before publication.

1. There are several inconsistencies between the introduction and findings that authors need to carefully review.

2. Make sure that your reference contains all the necessary details and PLOS ONE style.

3. Authors should acknowledge the limitation and elaborate the sampling at this time to revise the manuscript.

4. Decision: Major revision

. Please do not edit.

Reviewers' comments:

Reviewer's Responses to Questions

**Comments to the Author**

1. If the authors have adequately addressed your comments raised in a previous round of review and you feel that this manuscript is now acceptable for publication, you may indicate that here to bypass the “Comments to the Author” section, enter your conflict of interest statement in the “Confidential to Editor” section, and submit your "Accept" recommendation.

Reviewer #2: All comments have been addressed

Reviewer #3: All comments have been addressed

2. Is the manuscript technically sound, and do the data support the conclusions?

Reviewer #2: Yes

Reviewer #3: Yes

3. Has the statistical analysis been performed appropriately and rigorously? 

Reviewer #2: Yes

Reviewer #3: Yes

4. Have the authors made all data underlying the findings in their manuscript fully available?

Reviewer #2: Yes

Reviewer #3: Yes

5. Is the manuscript presented in an intelligible fashion and written in standard English?

Reviewer #2: Yes

Reviewer #3: Yes

6. Review Comments to the Author

Reviewer #2: (No Response)

Reviewer #3: The manuscript shows great improvements with its limitations, and I have a doubt on 2 points only.

1. How digitalization affect retention of mother? It might have effect on quality of service (e.g. timing, retraction of data, for data loss) with can affect the retention. However, How digitalization can directly have impact on maternal retention?

2. This is a quasi experimental study, and the authors report 100% response rate?

Could this be real in context of Ethiopia, specially in Amhara region with many conflicts and displaced populations?

Thank you!

7. PLOS authors have the option to publish the peer review history of their article (what does this mean? ). If published, this will include your full peer review and any attached files.

**Do you want your identity to be public for this peer review?** For information about this choice, including consent withdrawal, please see our Privacy Policy .

Reviewer #2: No

Reviewer #3: No

While revising your submission, please upload your figure files to the Preflight Analysis and Conversion Engine (PACE) digital diagnostic tool, https://pacev2.apexcovantage.com/ . PACE helps ensure that figures meet PLOS requirements. To use PACE, you must first register as a user. Registration is free. Then, login and navigate to the UPLOAD tab, where you will find detailed instructions on how to use the tool. If you encounter any issues or have any questions when using PACE, please email PLOS at figures@plos.org . Please note that supporting information files do not need this step.

---

## [Author Response · Author response to Decision Letter 1]

3 Feb 2025

Thank you so much dear esteemed editor for giving your precious time to review and provide very constructive comments. We have addressed all the comments from editor and reviewer as well. If anthing we missed during revision, please let us know to learn. Thank you again!

---

## [Editor Report · Decision Letter 2]

6 Feb 2025

Digitalizing the community health information system improved women’s retention on the maternal continuum of care pathway in northwest Ethiopia: a quasi-experimental study

PONE-D-24-09283R2

Dear author,

We’re pleased to inform you that your manuscript has been judged scientifically suitable for publication and will be formally accepted for publication once it meets all outstanding technical requirements.

Within one week, you’ll receive an email detailing the required amendments. When these have been addressed, you’ll receive a formal acceptance letter, and your manuscript will be scheduled for publication.

An invoice will be generated when your article is formally accepted. Please note that if your institution has a publishing partnership with PLOS and your article meets the relevant criteria, all or part of your publication costs will be covered. Please make sure your user information is up-to-date by logging into Editorial Manager at Editorial Manager®  and clicking the ‘Update My Information' link at the top of the page. If you have any questions relating to publication charges, please contact our Author Billing department directly at authorbilling@plos.org.

If your institution or institutions have a press office, please notify them about your upcoming paper to help maximize its impact. If they’ll be preparing press materials, please inform our press team as soon as possible—no later than 48 hours after receiving the formal acceptance. Your manuscript will remain under a strict press embargo until 2 pm Eastern Time on the date of publication. For more information, please contact onepress@plos.org.

Kind regards,

Birhan Tsegaw Taye

Academic Editor

PLOS ONE

Thank you for your careful consideration of the comments raised.

---

## [Editor Report · Acceptance letter]

PONE-D-24-09283R2

PLOS ONE

Dear Dr. Hailemariam,

I'm pleased to inform you that your manuscript has been deemed suitable for publication in PLOS ONE. Congratulations! Your manuscript is now being handed over to our production team.

Kind regards,

on behalf of

Mr. Birhan Tsegaw Taye

Academic Editor

PLOS ONE